# Protein Adhesives: Investigation of Factors Affecting Wet Strength of Alkaline Treated Proteins Crosslinked with Glyoxal

**DOI:** 10.3390/polym14204351

**Published:** 2022-10-15

**Authors:** Elena Averina, Johannes Konnerth, Hendrikus W. G. van Herwijnen

**Affiliations:** 1Institute of Wood Technology and Renewable Materials, Department of Material Sciences and Process Engineering, University of Natural Resources and Life Science, Konrad-Lorenz-Strasse 24, 3430 Tulln an der Donau, Austria; 2Wood K Plus—Kompetenzzentrum Holz GmbH, Altenberger Straße 69, 4040 Linz, Austria

**Keywords:** cross-linking agent, glyoxal, protein adhesives, wet strength, wood adhesives

## Abstract

Proteins obtained as side-products from starch production (potato and corn proteins) were investigated for wood adhesives application. To improve the wet strength of protein-based adhesives, glyoxal was added as a crosslinking agent. The effect of glyoxal on the wet strength of protein-based adhesives was investigated at different pH, protein: glyoxal ratios and solid content. The alkaline pretreatment of proteins was carried out by two different methods which reduced the molecular weight of proteins to different extents. The effect of molecular weight reduction on the wet strength of protein-glyoxal adhesives was also observed. It was found that pH level affects wet strength more significantly compared to solid content and protein-to-crosslinker ratio. Potato and corn proteins crosslinked with glyoxal showed maximal wet strength results in an acidic pH range

## 1. Introduction

Plant proteins are environmentally friendly and sustainable. Various industrial applications are exploiting biopolymers due to their characteristics, including wood adhesives [1,2]. Alternatives to fossil-based adhesives include wood adhesives made from plant proteins. Composite wood panels, such as plywood, particle boards and others commonly use adhesives with high adhesion strength and low costs, such as urea- and phenol-formaldehyde. The main source of these adhesives is limited and non-renewable. Formaldehyde-based resins may also emit carcinogenic formaldehyde during the manufacturing and use of panels [1,2,3]. Protein-based adhesives are becoming more popular as they are mainly based on renewable resources, thus reducing environmental impact [3,4,5].

Soybean proteins have been the most extensively studied ones for adhesive applications [6,7,8]. However, soy flour is mostly imported in Europe. Since the CO_2_ footprint of raw materials should be reduced, it would be favorable to grow plant proteins close to the market. Proteins from other plants, grown in Europe are, therefore, interesting to study [5,9].

In this work we compared alternative raw materials from plant sources abundant in Europe, namely corn, and potato protein concentrates, which were treated in alkaline conditions before crosslinking, to expose polar side groups of protein molecules [5,10], and investigated for their bonding performance and water resistance. These side streams are obtained from starch production as a by-product.

Poor water resistance is one of the main disadvantages of protein-based adhesives. A lot of research has been conducted to improve the wet strength of protein-based adhesives by adding active substances that crosslink protein functional groups [5,11,12]. Compared to formaldehyde, glyoxal is less toxic and less volatile [13]. Glyoxal is a bi-functional aldehyde, which is studied as a crosslinking agent for other bio-based adhesives, such as tannin [14,15] or lignin-based [16] ones. In the case of protein-based adhesives, however, glyoxal was studied mostly for soy protein-based adhesives in alkaline conditions [13,17,18]. In this work, we investigated a combination of factors, such as pH, degree of hydrolysis, solid content, and components ratio, which are assumed to affect the wet strength of protein-based adhesives to a different extent. Modified potato- and corn-protein glyoxal adhesives were synthesized and analyzed with the aim of improving the wet shear strength of beech veneer joints. 

## 2. Materials and Methods

Commercial protein concentrates were used in this work, namely corn protein concentrate (62% total protein content in dry substance which is 87%) and potato protein concentrate (80% total protein content in dry substance which is minimum 86%) were supplied by Agrana Research and Innovation Center GmbH (Tulln/Donau, Austria) and are further referred to as “proteins”. Aqueous glyoxal 40% solution of analytical grade (AR) was obtained from Carl Roth (Karlsruhe, Germany). Sodium hydroxide with 99% purity and formic acid 95% were bought from Sigma Aldrich (St. Louis, MO, USA). 

### 2.1. Solubility of Proteins 

Water soluble protein content was measured using Bicinchoninic Acid (BCA) protein assay kits [19,20]. Supernatant water-soluble protein content was determined with a BCA protein assay kit (Thermo Fisher Scientific, Waltham, MA, USA) with a fluorescence spectrophotometer (Fluostar Omega, Ortenberg, Germany) at a wavelength of 562 nm. Bovine serum albumin (BSA) was used as a standard (Sigma Chemical Co., St. Louis, MO, USA). The standard curve and results were calculated automatically via MARS Data analysis software (LabTech software, Tampa, FL, USA). The protein solubility was expressed as a percentage of the original nitrogen content of the sample.

### 2.2. Preparation of Adhesives

The alkaline treatment was performed according to two methods as follows. Figure 1 shows the steps for alkaline treatment for Method 1—portion-wise addition of protein and Method 2—one-time addition of protein followed by water elimination. The first method (method 1) was performed by portion-wise addition of protein concentrate to sodium hydroxide solution 1 M (80 mL). Sodium hydroxide solution was added to a three-necked flask equipped with a mechanical stirrer, a reflux condenser and a thermometer. The temperature was adjusted to 60 °C and protein concentrate (10 g) was added. After establishing a homogeneous mixture, the next portion (5 g) was added. The procedure was repeated until the amount of 35 g protein was added. The resulting solid content was 30%. For solids content of 20 and 25%, the amounts of 20 and 25 g of proteins were added.

The second method (method 2) was performed as follows. Sodium hydroxide 1 M (80 mL) was added to a three-necked flask equipped with a mechanical stirrer, a thermometer and a reflux condenser with a Dean–Stark trap. The temperature was adjusted to 60 °C and protein (10 g) was added and stirred while the evaporated water was condensed in a Dean–Stark trap. The resulting solid content was 30% after 195 min. The resulting solid content was measured by drying a 1 g sample of the alkaline protein mixture at 100 °C for 1 h and by calculating the average weight of three samples.

The resulting alkaline protein was placed in a beaker equipped with a mechanical stirrer and pH electrode and formic acid 60% (diluted from 95%) was added dropwise while stirring until the pH level shifted to 9, 6.5, or 4. After the addition of formic acid, the pH value was measured while stirring until a constant value was reached using a pH Meter (HI 2211, Hanna Instruments, Graz, Austria). After adjusting the pH to 4, protein mixtures were stirred for 30 min to obtain homogeneous adhesives based on potato protein (named pot) and corn protein (named corn). 

Glyoxal 40% water solution was added to protein adhesives in ratios protein to glyoxal of 2:1 for the first study of pH influence on wet strength and molecular weight distribution effect. For the investigation of three influencing factors, the ratios 1:1 and 5:1 were chosen as border levels with 2.5:1 as a central point. Ratios were calculated based on dry mass. Mixtures were stirred for 30 min until a homogeneous state was achieved to obtain protein-glyoxal adhesives named pot/gly and corn/gly. 

### 2.3. Viscosity

Viscosity measurements were carried out with a Modular Compact Rheometer MCR 302 (Anton Paar GmbH, Graz, Austria) using a cone-and-plate system with a diameter of 50 mm and a cone angle of 1° at a shear rate increasing from 100 1/s to 1000 1/s during 300 s with the viscosity value measured every 10 s at a temperature of 23 °C (EN 12092:2002). 

### 2.4. Tensile Shear Strength Testing by ABES

To determine the dry and wet tensile shear strength of the adhesives, an Automated Bonding Evaluation System (ABES, Adhesive Evaluation Systems, Inc., Corvallis, OR, USA) was used according to the standard ASTM D7998 [21]. Beech veneers with dimensions of 117 mm × 20 mm × 0.6 mm, stored beforehand in standardized climate conditions (20 ± 2 °C, 65 ± 4% relative humidity) for at least 7 days, were used as wood substrate. Adhesives were spread on the substrate at an amount of ~150 g/m^2^ (dry weight), with an overlap area of 20 mm × 5 mm and pressed at 110 °C for 5 min. Wet tensile shear strength was measured in a wet state after 1 h and 20 h water storage at room temperature. The samples were immersed in water immediately after the hot pressing. 

### 2.5. Reactivity Determination by DSC

The differential scanning calorimetry measurements were performed using a calorimeter type DSC 214-Polyma (Netzsch, Selb, Germany). Adhesive samples were heated from 20 °C to 200 °C at a heating rate of 10 K/min in high-pressurized steel crucibles. For data analysis, the software Netzsch-Proteus-80 was used.

### 2.6. Molecular Weight Determination 

Molecular weight distribution was measured by using polyacrylamide gel electrophoresis with sodium dodecyl sulfate (SDS-PAGE). A gradient gel 4–20% Mini-PROTEAN^®^ TGX (Bio-Rad, Hercules, CA, USA) was used to cover a broad range of molecular weights. The samples were incubated in an extraction buffer (0.25 M Tris, 1.92 M Glycine, 1% Dithiothreitol (DTT), and 0.1% SDS) overnight; afterward, the reduction of disulfide bonds was performed at 95 °C for 5 min. All samples were centrifuged at 4000 rpm for 10 min, and the supernatants were used to load the gels. The separating gel was run at a constant current of 30 mA for about 1.5 h. The gels were stained in Coomassie brilliant blue. Molecular weights of protein hydrolysates were estimated by using RotiVR-Mark 10–150 (Roth, Germany).

### 2.7. Experimental Design (DoE) and Statistical Analysis

Due to the high number of testing factors, a thoroughly planned experimental procedure was essential to optimize experimental efforts. A 2 k−p fractional factorial designed experiment is capable of reducing the number of samples to 10% of the full factorial design since the maximum and minimum levels were combined; based on this a characterization design of resolution V (26−1) was chosen (Box and Hunter 1961). Two repetitions (using five samples per repetition) of the 24 experimental combinations were performed; a center point, which combined all the mean levels, was repeated 10 times (using again five samples per repetition). The tensile shear strengths as results of each experiment were evaluated using the statistical software Design-Expert^®^ (version 10, Stat-Ease, Inc., Minneapolis, MN, USA). The arithmetic mean value was calculated for the five samples for each experimental combination. The responses and corresponding parameters were modeled and analyzed using an analysis of variances (ANOVA); additionally, a multivariate regression model was calculated [22]. The response surface method (RSM) generated maps of responses as three-dimensional plots. As a statistical method, RSM uses quantitative data from conducted experiments to determine regression model equations and derive operating conditions thereof. Direct experimental results were visualized using boxplots and SPSS Software (version 21, Statistical Package for Social Science, IBM^®^, Armonk, NY, USA). Values of experimental combinations with *n* = 5 samples (for each repetition) and *n* = 40 valid samples for the center point were compared. 

## 3. Results and Discussion

### 3.1. Effect of pH on Protein Solubility and Wet Strength of Protein/Glyoxal Adhesives

It is known that the pH of the media influences the charge of the protein [5,23]. At a pH level close to the isoelectric point (IP) proteins have zero charge or close to zero which makes them tend to precipitate. According to the literature, the isoelectric point of corn protein is close to pH 6.0–6.5 [24,25] and for potato protein close to pH 4.0–4.5 [26].

Figure 2a represents the solubility of potato and corn proteins as a function of pH. It was observed that the solubility of potato protein and corn proteins was at its lowest level at a pH close to the isoelectric point. The change in alkaline-treated proteins from a gel-like state with higher solubility at basic pH to a paste-like state with low solubility at a pH close to the isoelectric point can be seen in Figure 2b.

To study the possibility of using the protein at its lowest solubility for the reaction with glyoxal to improve wet strength, alkaline-treated potato and corn proteins shifted to different pH levels were mixed with glyoxal. The pH level of protein/glyoxal adhesive is lowered by the addition of glyoxal (from 9 to 8, or from 4 to 3); however, the pH level of protein before interaction with glyoxal is further referred to as “pH” in all cases for better comparability of the results with added and without added glyoxal. The protein solubility and agglomeration level before glyoxal is added affect the interaction with glyoxal.

It was observed that glyoxal reacts with potato and corn proteins added to alkaline treated proteins after pH was shifted to different values. The potato and corn adhesives with added glyoxal at a pH level higher than 11 were not applicable on wood substrates due to a dramatic viscosity increase (Figure 3b), which indicated that the reaction occurred in these conditions already at room temperature before hot pressing. Shifting the pH to lower levels reduced this effect. However, as can be seen in Figure 3a, the addition of glyoxal improves the wet strength of potato and corn proteins at different pH after the hot pressing to different degrees. The wet tensile shear strength in both cases of corn and potato proteins is substantially higher in the acidic range closer to the isoelectric point of proteins. It can be observed that low solubility correlates with higher wet strength performance for both proteins. Interactions between protein and glyoxal are mostly studied in alkaline conditions [13,17,18,27]. There are investigations on protein-glyoxal adhesives at different pH, which described that glyoxal reacts with soy protein in both alkaline and acidic conditions, preferably in alkaline conditions [13]. However, the possibility of protein crosslinking in the pH range close to the protein isoelectric point (IP) in the agglomerated state after preliminary unfolding by the alkaline treatment needs to be elaborate. The reaction between protein and glyoxal was studied earlier in soy flour in the acidic pH range between 1 and 3 [13], which is below IP and where the solubility of protein is higher than at IP [28]. Various further research has been conducted on glyoxal as a crosslinker for soy protein in alkaline conditions with additional components, such as urea or melamine [18,29]. The wet strength values achieved were below 1 Mpa; however, hot pressing temperatures below 140 °C were not investigated.

The energy release of potato-glyoxal and corn-glyoxal adhesives prepared at different pH derived from DSC measurements are plotted in Figure 4. The reaction peaks indicate that the reaction takes place in different pH ranges. The DSC analysis showed that the peak for pH 11 is less intense for both proteins and slightly shifted to a lower temperature, which may indicate that the reaction partly occurred before heating or is catalyzed by -OH groups. The onset for potato/glyoxal at pH 11 is shifted to a lower temperature compared to corn/glyoxal adhesive prepared at pH 11. The reason can be that potato protein contains more polar groups than corn protein, and therefore, more readily reacts with glyoxal [26,30]. The observed reaction peaks are assumed to represent only a part of the picture due to reactions partly occurring during the addition of the glyoxal to protein.

### 3.2. Effect of Molecular Weight Distribution on Wet Strength of Protein/Glyoxal Adhesives

Proteins treated in an alkaline solution by two different methods as described in Section 2.2 showed the influence of molecular weight distribution on wet strength. 

The molecular weight distribution (Figure 5a) of proteins was analyzed for hydrolyzed proteins prepared by portion-wise addition (method 1) and one-time addition (method 2). SDS measurements of the molecular weight of potato proteins pretreated with the portion-wise addition (method 1) show that the sample contains mainly protein molecules with a weight in the range of 15–20 kDa, while for the untreated potato protein, weights of up to 40 kDa could be observed. Although it must be considered that the molecular weight is reduced during the SDS-PADE method by sodium dodecyl sulfate, it still can be seen that after pretreatment with method 2 the molecular weight is more severely reduced. The SDS analysis of the corn protein sample pretreated with method 1 shows the presence of the molecules with the weight of 15 kDa, while after pretreatment with method 2 the molecules are evenly distributed in the gel volume and do not hold onto a certain position of the gel.

Analysis of viscosity as a function of shear rate also showed the difference between protein samples prepared by methods 1 and 2 (Figure 5b). Typically, protein alkaline blends exhibit non-Newtonian behavior, which is inherent in polymers [31]. All samples show decreased viscosity as the shear rate increases. Literature usually explains this behavior by breaking protein–protein interactions between proteins’ chains because of applied deformation [32,33,34]. Both potato and corn proteins show more pronounced non-Newtonian behavior after pretreatment with method 1, apparently due to the presence of longer molecules. The pH shift from basic to acidic conditions leads to the precipitation of protein which leads to the formation of a paste-like suspension with higher viscosity at a low shear rate. Different from protein samples pretreated with method 1, samples pretreated with method 2 do not show pronounced non-Newtonian behavior and almost do not change the viscosity with the change of shear rate, which is due to the smaller protein molecules.

The relation between wet strength and shear thinning behavior of protein-glyoxal adhesives prepared using protein samples pretreated with method 1 and method 2 was displayed in Figure 6. Viscosity as a function of shear rate for potato and corn proteins pretreated with methods 1 and 2 mixed with glyoxal at different pH and solid contents are displayed in Figure 6a. The viscosity profile for potato-glyoxal and corn-glyoxal adhesives repeats the trend of Figure 5b for proteins without glyoxal. The level of viscosity for proteins pretreated with method 1 crosslinked with glyoxal (Figure 6a) is higher than for the ones without glyoxal (Figure 5b). The level of viscosity for proteins pretreated with method 2 remains almost unchanged after the addition of glyoxal. We assume that due to the smaller molecules, the reaction with glyoxal does not lead to a significant network formation and to its interaction with the wood surface, which is essential for wood bonding. 

Figure 6b represents wet shear strength values for these samples, measured after storing glued wooden samples after hot pressing in water for 1 h. The wet strength is significantly lower for protein-glyoxal adhesives using protein pretreated by method 2 both for potato and corn proteins, which correlates with lower molecular weight (Figure 5a) and lower viscosity (Figure 5b and Figure 6a). Both potato and corn proteins show higher wet strength at higher solid content and lower pH.

It was already described in an example of soy protein that glyoxal is able to crosslink proteins [17,35] with the formation of Schiff-base structures (-C=N-) leading to improvement of wet strength. However, the effect of molecular weight distribution and severe molecule weight reduction on crosslinking with glyoxal and its influence on wet strength is not elaborate. Yue-Hong et al. [35] describe the positive effect of alkaline treatment and molecular weight reduction on adhesion properties and reduced viscosity. The negative effect of protein molecular weight reduction on wet strength as described in Yue-Hong et al. [35] is balanced by the addition of crosslinking agents. However, the present investigation shows that the crosslinking effect of glyoxal is relatively insignificant for proteins with excessively reduced molecular weight.

### 3.3. Effect of pH, Solids Content, and Protein/Glyoxal Ratio on Wet Strength of Protein/Glyoxal Adhesives

For the investigation of the net effect of three different parameters on the wet strength of potato-glyoxal and corn-glyoxal proteins, method 1 was chosen for the alkaline pretreatment of proteins. The used measured values were all combinations of the highest and lowest solid content combined with the highest and lowest amount of glyoxal at pH 4 and 9 and the center point corresponded to pH 6.5; the ratio glyoxal to protein 1 to 2.5 (corresponds to ratio 60% on the plot) and the solid content 25% (Figure 7). 

Figure 8 represents the graphical plots of the mathematical model using the measured values of wet strength after storing the glued wood samples in water for 20 h after the hot pressing. The 3D plots illustrate the effects of three factors (pH, amount of glyoxal and solid content) and their interactions on the tensile shear strength in wet conditions. Overall, potato protein-glyoxal adhesive shows better wet strength performance with maximum values of around 1.9 MPa compared to corn protein-glyoxal based adhesive with maximum values of around 1.4 MPa, most likely because of the higher content of polar groups in the potato protein amino acids content [26,36].

It was observed that pH affects the wet strength of potato-glyoxal and corn-glyoxal adhesives to a higher extent among the three investigated parameters. The highest wet strengths can be reached for both potato and corn proteins at pH 4 with a solid content of 30% and the ratio protein:glyoxal 1:1. However, at pH 4 the effect of the increase in glyoxal amount is less pronounced than at pH 9 both for potato and corn proteins. An increase in glyoxal amount at pH 9 leads to a wet strength improvement from 1 to 1.25 MPa for potato protein-based adhesive and from 0.5 to around 0.8 MPa for corn protein-based adhesive. When adding a higher amount of glyoxal to the protein mixture at pH 4, a change from 1.7 to 1.9 MPa in the case of potato protein and from 1.25 to 1.35 MPa in the case of corn protein is observed. 

Statistical evaluation was conducted with the help of analysis of variances (ANOVA) and multivariate data analysis. The adequacy of the model was justified with the correlation coefficient R^2^ value. The Predicted R^2^ for potato protein-based adhesives of 0.9518 is in reasonable agreement with the Adjusted R^2^ of 0.9639. Adequate precision measures the signal-to-noise ratio. A ratio greater than 4 is desirable. This model can be used to navigate the design space. Table 1 shows the impact of influencing factors on the response to potato protein/glyoxal adhesives. The probability value (*p*-value) of the model was highly significant. *p*-values greater than 0.05 indicate that the model term did not significantly influence the wet strength. The negative effect of pH means the increase in pH (to a more basic level) has a negative effect on wet strength.

Table 2 shows the impact of influencing factors on the response to corn protein/glyoxal adhesives. The probability value (*p*-value) of the model was highly significant. The Predicted R^2^ for corn protein/glyoxal adhesives of 0.9496 is in reasonable agreement with the Adjusted R^2^ of 0.9731; i.e., the difference is less than 0.2.

## 4. Conclusions

Potato protein-glyoxal and corn protein-glyoxal adhesives were investigated at different pH, protein:glyoxal ratios and solid content. Glyoxal significantly improved the wet bonding performance of the protein adhesives of veneer-based adhesive joints, which is indicated by the difference in wet strength between the protein with and without glyoxal. However, increasing the amount of glyoxal above the ratio protein: glyoxal 5:1 does not change the wet strength significantly at pH 4 for both proteins.

It was observed that pH affects the wet strength of potato-glyoxal and corn-glyoxal adhesives to the highest levels among the three investigated parameters. The alkaline pretreatment of proteins was carried out in two different ways which reduced the molecular weight of proteins to different extents. The effect of molecular weight distribution on the wet strength of protein-glyoxal adhesives showed that severe molecular weight reduction can significantly reduce the wet strength performance of protein-glyoxal adhesives and eliminate the positive effect of other parameters, such as pH, crosslinker amount and solid content.

Potato protein-glyoxal adhesive shows better performance than corn protein-glyoxal based adhesive, most likely because of the higher content of polar groups in potato protein amino acids groups. Potato protein-glyoxal adhesive prepared with the ratio protein:glyoxal 1:1 (dry weight), solid content 30% and pH 4 showed wet tensile shear strength close to 2 MPa after storing for 20 h in water. Corn protein-glyoxal adhesive prepared with the same parameters reaches wet strength of around 1.4 MPa after storing in the same conditions.

By adjusting the identified relevant parameters both potato and corn proteins are suitable to achieve suitable wet strength values and need to be further investigated for potential wood product applications.

## Figures and Tables

**Figure 1 polymers-14-04351-f001:**
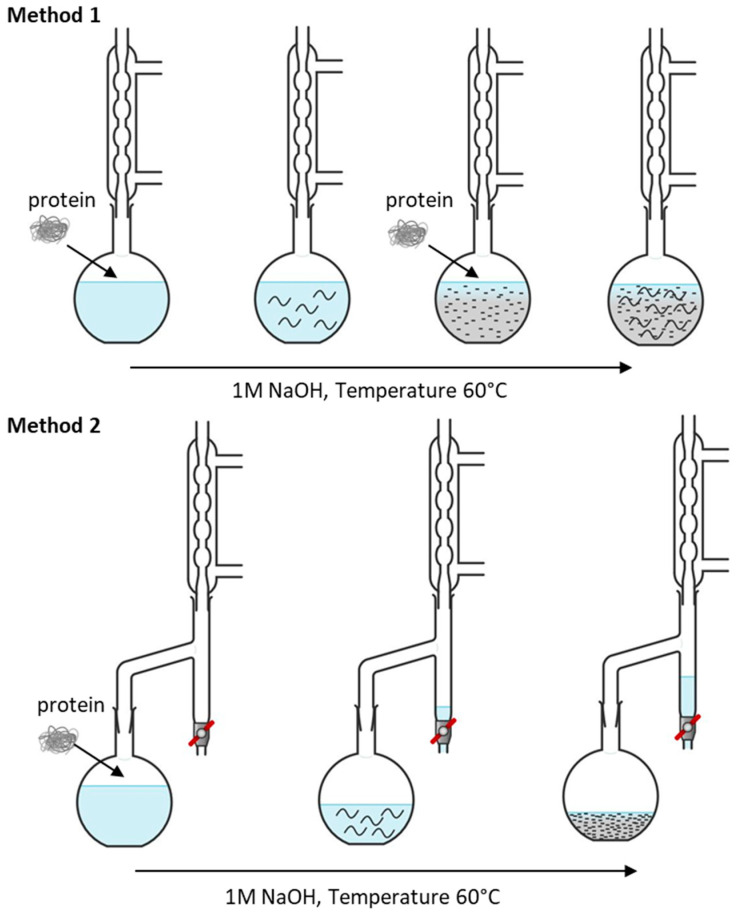
Schematic steps of protein alkaline treatment by Method 1 (portion wise) and Method 2 (one-time addition).

**Figure 2 polymers-14-04351-f002:**
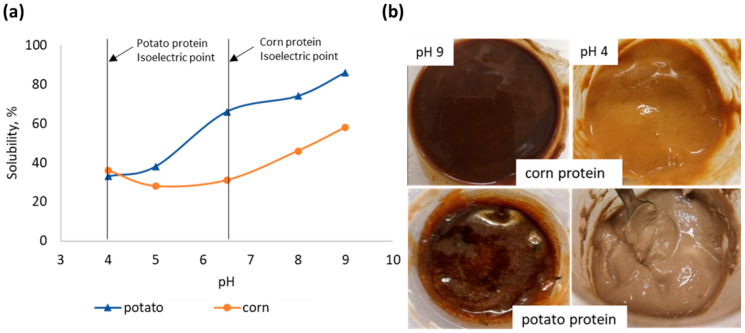
(**a**) Solubility of protein dispersions at different pH; (**b**) appearance of alkaline treated potato and corn proteins at a solid content of 25%.

**Figure 3 polymers-14-04351-f003:**
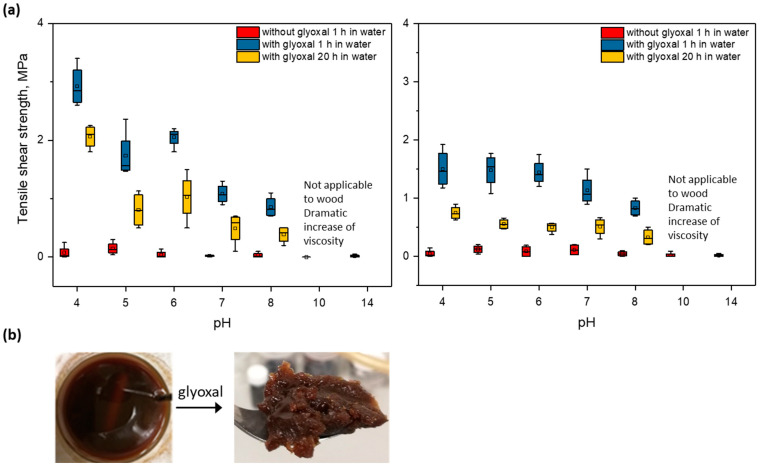
(**a**) tensile shear strength of potato/glyoxal (ratio 2:1 dry mass) and corn/glyoxal (ratio 2:1 dry mass) adhesives (method 1) at various pH levels; (**b**) increase in potato protein viscosity with solid content 25% after addition of glyoxal at pH 13.

**Figure 4 polymers-14-04351-f004:**
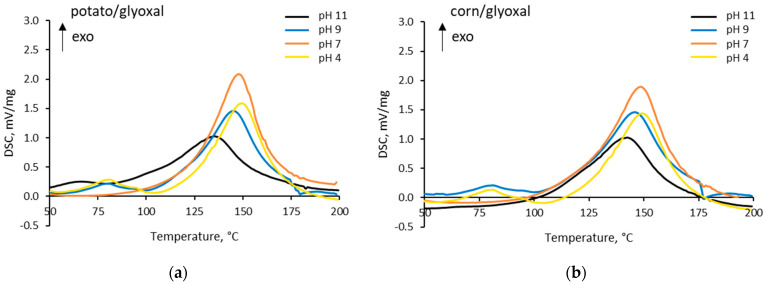
DSC graphs of (**a**) potato/glyoxal (ratio 2:1 dry mass) and (**b**) corn/glyoxal (ratio 2:1 dry mass) adhesives prepared at different pH levels (Method 1).

**Figure 5 polymers-14-04351-f005:**
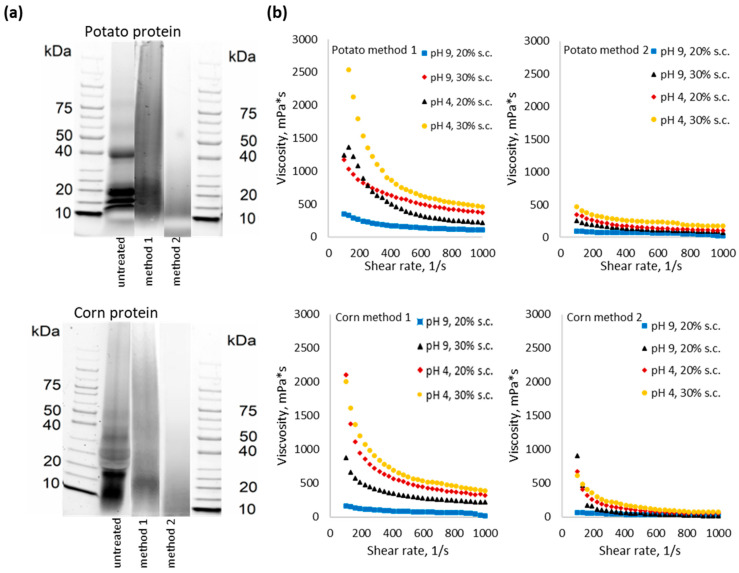
(**a**) Molecular weight distribution measured by SDS PAGE for differently prepared potato and corn proteins; (**b**) Viscosity (as a function of shear rate) of alkaline treated potato and corn proteins with pH shifted to different levels.

**Figure 6 polymers-14-04351-f006:**
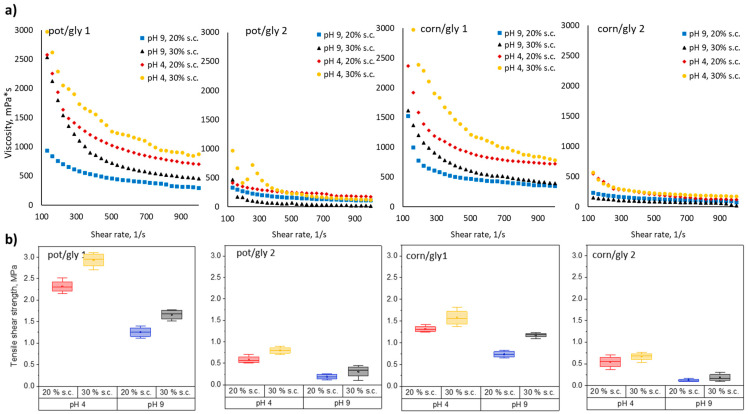
(**a**) Viscosity as a function of shear rate for potato-glyoxal (ratio 2:1 dry mass) and corn-glyoxal (ratio 2:1 dry mass) adhesives with solid content (s.c.) 20 and 30% using proteins pretreated with method 1 (pot/gly 1, corn/gly 1) and method 2 (pot/gly 2, corn/gly 2); (**b**) corresponding wet strength of protein/glyoxal and corn/glyoxal adhesives.

**Figure 7 polymers-14-04351-f007:**
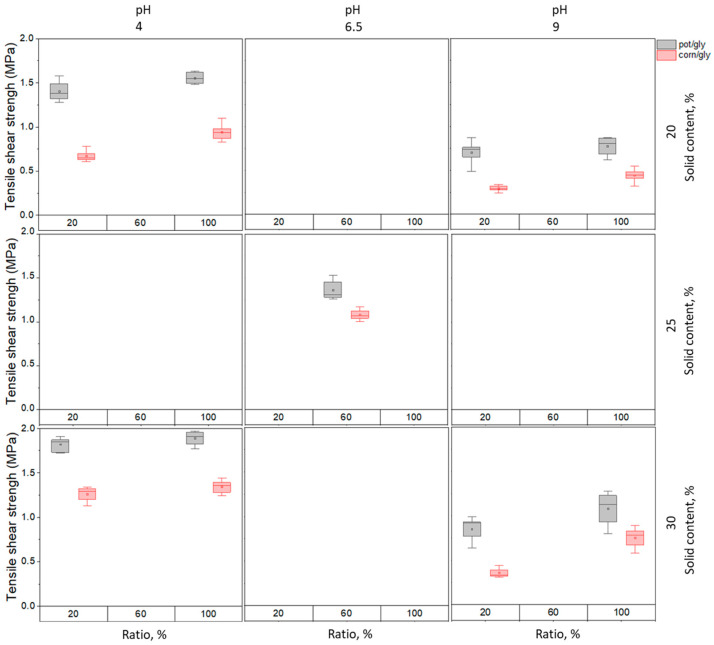
Tensile shear strength after 20 h water storage for potato protein/glyoxal adhesives (pot/gly) and corn protein/glyoxal adhesives (corn/gly) with glyoxal added at pH 4, 6.5 and 9 with ratios protein:glyoxal (dry) 1:1—100% on the scale, 2,5:1—60% on the scale.

**Figure 8 polymers-14-04351-f008:**
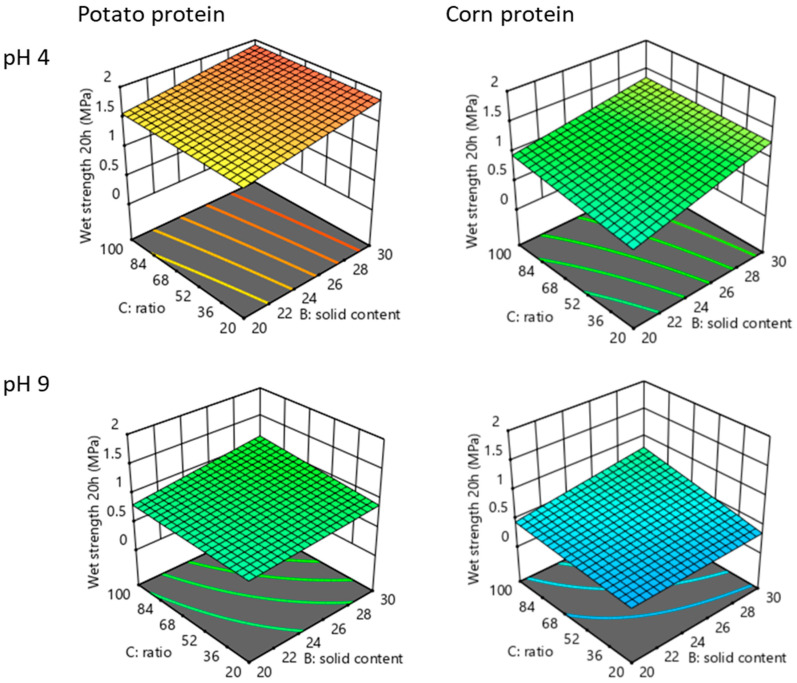
Three-dimensional model for the results of the experimental design of tensile shear strength (wet strength) in wet conditions after 20 h of soaking in water for potato/glyoxal and corn/glyoxal ratios (dry mass) adhesives. C = 100 corresponds to the ratio glyoxal:protein 1:1; C = 60 corresponds to the ratio glyoxal:protein 1:2.5; C = 20 corresponds to the ratio glyoxal:protein 1:5.

**Table 1 polymers-14-04351-t001:** Influencing factors and their interactions of processing parameters on wet strength after 20 h in water for potato protein/glyoxal adhesives.

Parameter		Final Equation in Terms of Coded Factors	*p*-Value		% Contribution
	Model	+1.29	<0.0001	significant	83.73
A	pH	−0.4188	<0.0001	significant	11.11
B	Solid content, %	+0.1525	<0.0001	significant	1.51
C	Ratio, %	+0.0562	0.0102	significant	0.5044
AB	Interaction A × B	−0.0325	0.1088		0.2694
AC	Interaction A × C	+0.0238	0.2312		0.1074
BC	Interaction B × C	+0.0150	0.4424		0.3611
ABC	Interaction A × B × C	+0.0275	0.1693		83.73

**Table 2 polymers-14-04351-t002:** Influencing factors and their interactions of processing parameters on wet strength after 20 h in water for corn protein/glyoxal adhesives.

Parameter		Final Equation in Terms of Coded Factors	*p*-Value		% Contribution
	Model	+0.7688	<0.0001	significant	59.47
A	pH	−0.2938	<0.0001	significant	22.02
B	Solid content, %	+0.1787	<0.0001	significant	9.72
C	Ratio, %	+0.1187	<0.0001	significant	3.75
AB	Interaction A × B	−0.0737	<0.0001	significant	0.5697
AC	Interaction A × C	+0.0288	0.0533		0.0872
BC	Interaction B × C	+0.0113	0.4228		2.59
ABC	Interaction A × B × C	+0.0612	0.0005		59.47

## Data Availability

Not applicable.

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
