# Peer review of "Protein Adhesives: Investigation of Factors Affecting Wet Strength of Alkaline Treated Proteins Crosslinked with Glyoxal"

_polymers, 2022, doi:10.3390/polym14204351_

Round 1
Reviewer 1 Report
Manuscript is prepated in good quality and methods are adequately described, as well as conclusions responds to investigated objectives. There are no much to correct, still there can be improved several shortcomings:
1.The abbreviation of the term "isoelectric point" starts too late at the section 3.1. The full name ar named about 5 times before the abbreviation.
2. In section 3.3. (Fig. 7 and 8) more detailed interpretation of the results would give better understanding of the plots. By mentioning some important values of the best or worst results.
Author Response
Dear Reviewer!
Thank you for your comments and suggestions, which helped us to improve our paper. Please find our answers in the text below in bold and the modifications in the paper in red font.
Best regards,
Elena Averina and co-authors

Reviewer 2 Report
The organization and presentation of this manuscript are good. The research content is also suitable for publication in this journal, and the research results are worthy of recognition. I only put forward one question for the author to consider: when protein adhesives is prepared, whether all reaction products were detected? After the material is actually used, are there any other volatile substances released after a period of time? For adhesives, the release of formaldehyde is usually one of the most concerned problems of consumers.
Author Response

(The authors gave the same response as above.)

Reviewer 3 Report
The review of a manuscript „Protein Adhesives: Investigation of factors affecting wet strength of alkaline treated proteins crosslinked with glyoxal” submitted by Averina et al. to Polymers Journal.
The subject of the article is in line with the research trend regarding adhesives based on vegetable proteins. I believe that trying to use non-soy protein is fully justified. Abstract is informative, it could be more specific in terms of the results, however, I understand that in case of such inclusive research it is hard to fully present the results in the abstract and don’t exceed the limited number of characters. The keywords should be listed in alphabetical order. Introduction part is rather short. Maybe authors should consider to include the description of more results of studies focused on gloxal-crosslinked protein adhesives. For now it contains mostly the general information. I have some suggestions: “plywood, composite wood panels and furniture production ….” Plywood is a composite wood panel and it is used in furniture production, it should be written in different way. Second paragraph contains interesting information and I understand the meaning of it but it should be rewritten to be more clear too. Materials and Methods section is presented rather clearly and it shows a high scientific soundness. Methods are selected properly. What was the dry substance in this commercially available concentrates? BCA abbreviation needs to be explained. Why is Figure 4 that far in the text from methodology part? Is there a reason of using such thin veneers? Could Authors provide the details about the shear strength testing, the samples dimensions etc.? Is it done following any standards? The results and discussion section is presented clearly, the results are usually compared with other findings which is a crucial element of scientific work. I can find the explanations for almost every phenomenon which was found in this study. Page 4, “however, the pH level of protein before interaction with glyoxal is important”, can you please provide a more extensive explanation why at this point it is important since the incorporation of crosslinker affected the pH. Page 7 “pronounced non-newtonian behavior and almost do not change the viscosity” can authors explain what is the meaning of it in terms of wood bonding? The graphs and statistical evaluation are correct and interesting. The conclusions are more like a summary and it concerns the most important aspects of the study.
In general, in my opinion this study could be a valuable reference for studies on plant-based protein adhesive. I appreciate the effort of authors and I hope my suggestion will contribute to the improvement of the paper.
Author Response

(The authors gave the same response as above.)
